# Repeated Measures Regression in Laboratory, Clinical and Environmental Research: Common Misconceptions in the Matter of Different Within- and Between-Subject Slopes

**DOI:** 10.3390/ijerph16030504

**Published:** 2019-02-11

**Authors:** Donald R. Hoover, Qiuhu Shi, Igor Burstyn, Kathryn Anastos

**Affiliations:** 1Department of Statistics and Biostatistics and Institute for Health, Health Care Policy and Aging Research, Rutgers University, Piscataway, NJ 08854, USA; 2School of Health Sciences and Practice, New York Medical College, Valhalla, NY 10595, USA; Qiuhu_shi@nymc.edu; 3Environmental and Occupational Health Dornsife School of Public Health, Philadelphia, PA 19104, USA; igor.burstyn@drexel.edu; 4Albert Einstein College of Medicine, Montefiore Medical Center, Bronx, NY 10467, USA; kanastos@montefiore.org

**Keywords:** within-/between-subject associations, repeated measures, cross-sectional regression, generalized estimating equations, mixed models, working correlation structure

## Abstract

When using repeated measures linear regression models to make causal inference in laboratory, clinical and environmental research, it is typically assumed that the within-subject association of differences (or changes) in predictor variable values across replicates is the same as the between-subject association of differences in those predictor variable values. However, this is often false. For example, with body weight as the predictor variable and blood cholesterol (which increases with higher body fat) as the outcome: (i) a 10-lb. weight increase in the same adult affects more greatly an increase in cholesterol in that adult than does (ii) one adult weighing 10 lbs. more than a second indicate higher cholesterol in the heavier adult. A 10-lb. weight gain in the first adult more likely reflects a build-up of body fat in that person, while a second person being 10 lbs. heavier than the first could be influenced by other factors, such as the second person being taller. Hence, to make causal inferences, different within- and between-subject slopes should be separately modeled. A related misconception commonly made using generalized estimation equations (GEE) and mixed models on repeated measures (i.e., for fitting cross-sectional regression) is that the working correlation structure only influences variance of the parameter estimates. However, only independence working correlation guarantees that the modeled parameters have interpretability. We illustrate this with an example where changing the working correlation from independence to equicorrelation qualitatively biases parameters of GEE models and show that this happens because within- and between-subject slopes for the outcomes regressed on the predictor variables differ. We then systematically describe several common mechanisms that cause within- and between-subject slopes to differ: change effects, lag/reverse-lag and spillover causality, shared within-subject measurement bias or confounding, and predictor variable measurement error. The misconceptions we describe should be better publicized. Repeated measures analyses should compare within- and between-subject slopes of predictors and when they do differ, investigate the causal reasons for this.

## 1. Introduction

We focus on two common misconceptions that are made in research while fitting repeated measures regression with generalized estimating equations (GEE) and mixed models (MM). Misconception-A: The association between the predictor variable and the outcome across different measures from the same subject (within-subject) is the same as the association of that variable with the outcome between measures from different subjects (between-subject). In fact, these associations often differ, which should be considered when making causal inference. For example, consider weight as the predictor and cholesterol the outcome given the well-known association of higher serum cholesterol and with greater body fat: (i) a 10 lb. increase in the same adult more likely indicates greater difference in serum cholesterol than does (ii) one adult being 10 lbs heavier than a second adult. A 10-pound weight gain in the same adult more likely reflects a build-up of body fat in that person, while the first adult being 10 pounds heavier than the second could be influenced by other factors such as the first adult being taller than the second. Misconception-B: The working correlation structure used in GEE and MM models is only a nuisance factor that impacts precision of model parameter estimates. As illustrated and explained in the next Section (and Table 1), the wrong choice for working correlation structure biases parameter estimates. 

Both of these misconceptions are related, but the analytical details are complicated. To explore this further, Section 2 begins with an illustration of Misconception-B in real data. Section 2 also explains how this relates to Misconception-A and why independence working correlation must be used for creation of predictive models using “cross-sectional regression” on repeated measures. Then, Section 3 details how separation into within- and between-subject associations is needed for using repeated measures regression to makes causal inference. Section 4 describes epidemiological mechanisms that can cause within- and between-subject slopes to differ. Section 6 summarizes and explores further implications for statistical practice in applied research.

## 2. Cross-Sectional and Between/Within-Subject Linear Models with Repeated Measures

We begin here with some notation. Consider repeated measures on *n* subjects denoted by *i = 1,2, …, n*. The “subjects” can either be persons with longitudinal repeated measures, or, as is common in environmental epidemiology, can be cities, schools, neighborhoods, census tracks, hospitals, etc. Each subject has *J_i_* different observations enumerated by *j* = *1*, …, *J_i_*. For example, these *J_i_* different observations could be taken at times *t_i1_* < *t_i2_* <… < *t_iJi_*, on the same person when the “subject” is a person or from *J_i_* different persons living in the same neighborhood when the “subject” is a neighborhood. For *J_i_* constant across *i*, (i.e., always the same number of repeat measures for a subject), we drop the *“i”* subscript and denote *J*. Let us consider that the observations have continuous outcomes *Y_ij_* and *K* predictor (or exposure) variables ${\underset{˜}{X}}_{ij}=X_{1,ij},X_{2,ij},\ldots ,X_{K,ij}$. When K = 1, we drop the “K” enumeration, using *X_ij_* for the only predictor. Linear regression models for E[*Y_ij_|*
${\underset{˜}{X}}_{ij}$] or E[*Y_ij_|X_ij_*] are fit in the analyses described here. However, the overall conclusions we obtain on these linear regression models can be generalized to discrete outcomes (i.e., logistic regression) and survival analyses. 

**2A Cross-Sectional (CS) Regression.** The most commonly fitted linear regression model on repeated measures does not separate within- and between-subject associations and is usually written out as *Y_ij_ = α + β_1_X_1,ij_ + β_2_X_2,ij_ + … + β_K_X_K,ij_* + *ε_ij_*. This is denoted as “cross-sectional (CS) regression” particularly for longitudinal repeated measures. We add a subscripted “CS” to the β’s to distinguish these slopes from between-subject (BS) and within-subject (WS) slopes defined in Section 2B. The CS regression model here is thus denoted as
(1)$$Y_{ij}={\underset{˜}{\beta }}_{CS}{\underset{˜}{X}}_{ij}=\alpha_{CS}+\beta_{1,CS}X_{1,ij}+\beta_{2,CS}X_{2,ij}+\ldots +B_{K,CS}X_{K,ij}+\varepsilon_{ij}$$
where $\alpha_{CS},\;\beta_{1,CS},\;\beta_{2,CS},\;\ldots ,B_{K,CS}$ are parameters (fixed effects), while *ε_ij_* is error with E[ε_ij_] = 0 that is independent between different subjects *i* and *i’*, but may be correlated for *j ≠ j’* within the same subject. It should be noted that the intercept is fixed at the same $\alpha_{CS}$ for each subject. Should the actual intercepts differ between subjects (i.e., be $\alpha_{CS,i}$) as random intercepts, then for both MM and GEE, the difference $\alpha_{CS,i}-\alpha_{CS}$ is incorporated into the error term *ε_ij_* of (1) and the within-subject correlation of that error [1]. Using (vs. not using) random intercepts does not influence the point estimates of $\alpha_{CS},\;\beta_{1,CS},\;\beta_{2,CS},\;\ldots ,B_{K,CS}$ or the variance of these estimates for mixed models [1]. However, for GEE using a different intercept on each subject (with each intercept now adding a new parameter) creates too many parameters for the asymptotic properties of GEE model to hold in our examples (and in general) which destabilizes parameter estimates [2].

Again for K = 1, the subscript for K is dropped and the model is $Y_{ij}=\alpha_{CS}+\beta_{CS}X_{ij}$ + *ε_ij_*. The main goal of CS regression is to first obtain estimates ${\underset{˜}{\hat{\beta }}}_{CS}$ for ${\underset{˜}{\beta }}_{CS}$ and then input ${\underset{˜}{\hat{\beta }}}_{CS}$ into (1) in order to estimate future unobserved Y’s from observed ${\underset{˜}{X}}_{ij}$’s as ${\underset{˜}{\beta }}_{CS}{\underset{˜}{X}}_{ij}$. Cross-sectional regression is also used to make adjusted (causal) inference on the covariate associations in ${\underset{˜}{\hat{\beta }}}_{CS}$, but, as we show later, doing this may be problematic. 

Table 1 presents parameter estimates from repeated measure cross-sectional regression (1) to a clinical measure of glomerular filtration rate (EGFR) from the Modification of Diet in Renal Disease Study (MDRD) [3]. Formula (1) with EGFR as the outcome Y and three predictor variables (X_1_, X_2_, X_3_) = **(**HIV infection, serum albumin, blood urea nitrogen (BUN)) was fit to 10,782 semi-annual measures of 584 women at the Bronx-site of the Women’s Interagency HIV Study (WIHS) [4]. Higher EGFR values indicate better renal function. The models assume that the within- and between-subject associations of the predictor variables are the same. We later show this assumption is incorrect. The parameter estimates of Table 1 were calculated using GEE [1] with both independence (GEE-IND) in columns 2–4 and equicorrelation (GEE-E) columns 5–7 for the working correlation structure of model residuals from repeated measures in the same person. We again note that this model (1) has a fixed intercept across all subjects with the error term being independent between different subjects. However, otherwise in Table 1 (and elsewhere in the paper) the within subject correlation structure of the error is allowed to be either (i) independent within the same subject (GEE-IND) or (ii) to have the same correlation for all outcomes within the same subject (GEE-E). The second condition (i.e., equicorrelation) is equivalent to fitting a random subject intercept model [1].

Most of today’s literature providing guidance on fitting repeated measures linear regression (i.e., [5,6,7,8,9,10,11,12,13]) qualitatively describes working correlation as a “nuisance factor” that does not alter model parameters and states that “*the working correlation that minimizes variance of parameter estimates should be chosen”*. However, in Table 1, the parameter estimates for BUN (per g/dL), from GEE-E, of −1.22; 95% confidence interval (CI) (−1.46, −0.99) is both qualitatively and statistically higher than the corresponding GEE-IND estimate of −1.87; 95% CI (−2.12, −1.62). For HIV, the parameter estimates of −3.86, *p* = 0.0081 from GEE-E is qualitatively lower than that from GEE-IND −2.04 and *p* = 0.19. Clearly, changing the working correlation from independence to equicorrelation qualitatively and statistically changes the parameter estimates. Thus, this correlation structure is not a nuisance factor.

When faced with such a dilemma of qualitatively and statistically different parameter estimates from the same model fit to the same data with only the working correlation structure changed (as is shown in Table 1), investigators typically go to published guidance on which correlation structure to use. To that end, based on the within-subject correlation of residuals being 0.45 in GEE-E (and in MM-E), and the quasi-likelihood independence criteria goodness of fit statistic (QIC) = 10,836.27 for GEE-E being smaller than the QIC = 10,847.14 for GEE-IND (or the Akaike information criteria goodness of fit statistic (AIC) from a mixed model using equicorrelation (MM-E) of (AIC = 94,934.5) being smaller than AIC = 99,374.5 from a mixed model using independence (MM-IND) as shown in Table A1 in Appendix A), almost all articles providing model fitting guidance [5,6,7,8,9,10,11,12,13] point towards using equicorrelation as the working correlation structure. However, as the rest of Section 2 describes in detail, this guidance is problematic as only the parameter estimates obtained by using independence working correlation can have any meaning for cross-sectional regression.

But first we make two brief asides. First, we note that if MM, rather than GEE are used for Table 1, the corresponding parameter point estimates in Table 1 using independence correlation (MM-IND) and equicorrelation (MM-E) are essentially unchanged [1]. (See Appendix A for details on parameter estimates from MM fit to this data with independence and equicorrelation correlations structures). However, due to non-robustness of MM, GEE is preferable for this specific example. Second, we note that the differences observed in Table 1 occur not only between independence and equicorrelation. Any different choice of correlation structure, such as AR(1), Toeplitz, unstructured, etc. will result in different parameter estimates (results not shown). For simplicity, we focus this article on only two structures: independence and equicorrelation. 

**2B Between-/Within-Subject Slope (BS/WS) Regression.** While investigators almost never consider this in practice, it has long been noted that slopes on changes of *X_ij_* within the same subject *i* differ from cross-sectional slopes on between subject-measure differences in *X_ij_* [14,15,16,17]. To illustrate this, consider the cross-sectional model of a laboratory measure cholesterol (*Y_ij_*), which is well known to be higher in people with more body fat. To that end, the predictor is body weight (*X_ij_*) with *E*[*Y_ij_*] = $\alpha_{CS}+\;\beta_{CS}X_{ij}$. As described in the Introduction, the cross-sectional slope *β_CS_* for association of a 10 lbs. weight difference between two different adults for cholesterol is less than the slope for association of a 10 lbs. within-subject weight change for the same adult on cholesterol, which we denote as *β_WS_*. Again, the reason *β_CS_* is less than *β_WS_* is that: (i) a 10 lb. cross-sectional weight difference between two adults often reflects greater height in one of the persons, but (ii) a 10 lbs. weight increase in the same adult is not influenced by height difference and thus is more likely due to more body fat after the 10-lbs weight gain. Thus, since greater body fat is what is directly associated with more cholesterol, the within-person association of a 10-lb. weight increase with cholesterol is greater than the cross-sectional repeated measures association with a 10-lb. weight difference between two persons. 

Common within-person height creates a shared within-subject measurement bias from this extraneous factor for subject *i* (denoted *E_i_*) on weight as a predictor of cholesterol. To that end, many investigators adjust weight for height using body mass index = *wt/ht^2^* to remove this effect of height on weight. As Figure 1a illustrates, if *TX_ij_* = body mass index (*wt/ht^2^*) were the true predictor of *Y_ij_*, and *H_i_* = height (which does not change with *j* in the same *i*), then *X_ij_ = TX_ij_ * (H_i_)^2^* contains this shared within-subject measurement bias from common *H_i_* which again we denote as *E_i_* in Table 1a to confer it is an extraneous within-subject bias. Section 4 describes more settings where $\beta_{WS}\neq \beta_{CS}$.

While for weight it is possible to remove the common shared within-subject bias from height by dividing by *ht^2^*, this is not the case for less well-understood causal relationships. Therefore, to model and account for a bias such as this, linear regression models fit for making causal inference can decompose the associations into “within-subject” slopes (${\underset{˜}{\beta }}_{WS}$), described above, and “between-subject” slopes (${\underset{˜}{\beta }}_{BS}$), described below, which capture associations of subjects’ central tendencies of the exposure. To do this, subject means of the predictor variables ${\underset{˜}{\bar{x}}}_{i}={\bar{x}}_{1,i},{\bar{x}}_{2,i},\ldots ,{\bar{x}}_{K,i}$ are calculated, where ${\bar{x}}_{k,i}=\sum_{j=i}^{Ji}X_{k,ii}/J_{i}$. Then *Y_ij_* is modeled as a combination of “between-subject” slopes from ${\bar{x}}_{k,i}$ (that could be influenced by the common person measurement bias in Figure 1) and “within-subject” slopes from deviations of *X_k,ij_* about ${\bar{x}}_{k,i}$ which will be free of such a bias, since the comparison is within person.
(2)$$\begin{array}{c} Y_{ij}=\alpha_{BS/WS}+\beta_{1,BS}{\bar{x}}_{1,i}+\beta_{2,BS}{\bar{x}}_{2,i}+\ldots +\beta_{K,BS}{\bar{x}}_{K,i}+\\ +\beta_{1,WS}(X_{1,ij}-{\bar{x}}_{1,i})+\beta_{2,WS}(X_{2,ij}-{\bar{x}}_{2,i})+\ldots +\beta_{K,WS}(X_{K,ij}-{\bar{x}}_{K,i})+\varepsilon_{ij} \end{array}$$

As described for (1), this is a fixed intercept model that is functionally equivalent to a random intercept model for MM. When K = 1, we have $Y_{ij}=\alpha_{BS/WS}+\beta_{BS}{\bar{x}}_{i}+\beta_{WS}(X_{ij}-{\bar{x}}_{i})+\varepsilon_{ij}$. To illustrate this for our earlier example with Y_ij_ = cholesterol and X_ij_ = weight, let $\alpha_{BS/WS}=30$, *β_BS_* = 0.9 and *β_WS_* = 3, such that $Y_{ij}=30+0.9{\bar{x}}_{i}+3(X_{ij}-{\bar{x}}_{i})+\varepsilon_{ij}$. If person *i* had an average value of ${\bar{x}}_{i}$ = 210 across all *J_i_* measures with the *j^th^* measure being *X_ij_* = 200, then for the person-visit at time *t_ij_*, E[Y_ij_] = 30 + 0.9(210) + 3(200-210) = 189. 

Now we make some technical asides. First, the choice of the observed ${\bar{x}}_{k,i}$ as the “central tendency” of *X_k,ij_* for subject *i* is necessary as $\mu_{k,i}$ a person’s “true average weight” over the entire time period is unknown, but for *J_i_* large enough, ${\bar{x}}_{k,i}$ should be close to $\mu_{k,i}$. Thus, while *β_k,WS_* only captures association with within-subject change in *X_k,ij_, β_k,BS_* inherently contains some *β_k,WS_* from deviation of (${\bar{x}}_{k,i}-\mu_{k,i}$); especially for small *J_i_*. This situation is described for occupational epidemiology research, where often an average of personal exposure measurements is computed as estimate of true exposure of a “subject”, defined as either an individual, or group of individuals that share a job [18]. Second, the implicit assumption that *β_k,WS_* is well defined may also not always be true. For example, “*β_k,WS_*” could differ by time separation t_ij_ – ti_j’_. Perhaps for *k* = weight, a weight gain of 10 lbs. in one month creates a shock that hyper-elevates cholesterol, but a 10 lbs. weight gain over 12 months does not, in which case $\beta_{k,WS}|(t_{ij}-t_{ij\prime })=1>\beta_{k,WS}|(t_{ij}-t_{ij\prime })=12$. Third, if the investigator is only interested in the within-subject slopes he/she can substitute as a *fixed **effect* a different subject intercept $\alpha_{WS,i}$ for the between-subject slopes in (2) with the model reducing to $Y_{ij}=\alpha_{BS,i}+\beta_{1,WS}(X_{1,ij}-{\bar{x}}_{1,i})+\beta_{2,WS}(X_{2,ij}-{\bar{x}}_{2,i})+\ldots +\beta_{K,WS}(X_{K,ij}-{\bar{x}}_{K,i})+\varepsilon_{ij}$. 

Despite these technical caveats, the within- vs. between-subject decomposition in (2) is used to test whether $\beta_{k.BS}=\beta_{k.WS}$ so that, as shown in Section 2C, they also equal $\beta_{k.CS}$ and thus the separated *WS* vs. *BS* decomposition can be collapsed to (1). Due to the orthogonal decomposition of *X_k,ij_* about ${\bar{x}}_{k,i}$ this previous test for collapsing the within- vs. between-subject decomposition is a two-sample z-test of parameter estimates from fitted models comparing $|{\hat{\beta }}_{k,BS}-{\hat{\beta }}_{k,WS}|/\sqrt{Var({\hat{\beta }}_{k,BS})+Var({\hat{\beta }}_{k,WS})}$ to *Z_1-α/2_* [17]. The within- vs. between-subject decomposition is mostly used for inference on adjusted (causal) associations of the *X_k,ij_’s* on *Y_ij_’s*. It is typically not used to produce models to estimate future unknown *Y_ij_* from known ${\underset{˜}{X}}_{ij}$ as such estimation often only happens in settings where just one observation per subject is available, hence $X_{k,ij}\equiv {\bar{x}}_{k,i}$.

We refit the analyses of Table 1 to illustrate that the impact of choice of correlation structure (i.e., GEE-IND vs. GEE-E working correlation structure) is eliminated in our example after making a within- vs. between-subject decomposition. Please note that there were no new HIV infections after study entry; so $X_{HIV,ij}\equiv {\bar{x}}_{HIV,i}$ meaning that the within-subject association of change of HIV infection status cannot be modeled. For within-subject associations of BUN and albumin, GEE-IND and GEE-E gave identical point estimates, because centering about ${\bar{x}}_{k,i}$ makes comparisons entirely within-subject and invariant to these correlation structure choices (although within-subject estimates could differ slightly if autoregressive (AR (1)) or other formulations for intra-subject correlation of residuals had been used). There were small GEE-IND vs. GEE-E differences on the between-subject slopes as was observed elsewhere [19]. For example, the point estimate for between-subject HIV status is −1.16; 95% CI (−4.21, 1.88) in the GEE-IND of Table 2 versus −1.57; 95% CI (−4.47, 1.33) with GEE-E.

From now on, we only examine GEE-IND results for within- between-subject decomposition models, as GEE-E results are similar. For BUN and GEE-IND, the within-subject ${\hat{\beta }}_{BUN,WS}$ = −1.11, 95% CI (−1.34, −0.88) is qualitatively and statistically closer to 0 than is the corresponding between-subject slope ${\hat{\beta }}_{BUN,BS}$ = −2.72, 95% CI (−3.10, −2.33). However, serum albumin goes the other way: the within-subject slope ${\hat{\beta }}_{ALB,WS}$ = −10.70, 95% CI (−12.99, −8.40) is statistically further from 0 than is the corresponding between-subject GEE-IND ${\hat{\beta }}_{ALB,BS}$ = −3.27 with a 95% CI (−7.88, 1.33) that overlaps 0. The QIC is lower (10,857.62) for equicorrelation than for independence (10,866.64) which perhaps now indicates an advantage to the former correlation structure in this setting where the slopes have been correctly decomposed.

One might wonder how to interpret differences in the within- and between-subject slopes for causal inference, including the reasons that these slopes were different? This in part will depend on the hypotheses of interest (and we did not have any for this illustrative example). However, general rules also apply, although we are unaware of any systematic exploration of reasons why the between-subject slopes ${\underset{˜}{\beta }}_{BS}$ (or ***β_BS_*** for *K* = 1) could differ from within-subject slopes ${\underset{˜}{\beta }}_{WS}$ (or ***β_WS_*** for *K* = 1). and the resultant implications for causal inference. Before outlining these rules, it is important to note an important relationship among cross-sectional, within-subject and between-subject slopes.

**2C Relationship between**
${\underset{˜}{\beta }}_{CS}$**,**
${\underset{˜}{\beta }}_{WS}$
**and**
${\underset{˜}{\beta }}_{WS}$. Now ${\underset{˜}{\beta }}_{CS}$ averages ${\underset{˜}{\beta }}_{WS}$ and ${\underset{˜}{\beta }}_{BS}$ according to relative variances of the subject means (i.e., the ${\underset{˜}{\bar{x}}}_{i}$) vs. the variance of the repeated measures about those sample means (i.e., the ${\underset{˜}{X}}_{i}-{\underset{˜}{\bar{x}}}_{i}$) [17]. For example, with *K* = 1, if $\sigma_{\bar{x}}^{2}$ is the population variance of the within-person mean ${\bar{x}}_{i}$ and $\sigma_{X-\bar{x}}^{2}$ is the population variance of the deviations of differences of the repeat measures *X_ij_* from their ${\underset{˜}{\bar{x}}}_{i}$, then
(3)$$\beta_{CS}=\beta_{BS}\sigma_{\bar{x}}^{2}/(\sigma_{X-\bar{x}}^{2}+\sigma_{\bar{x}}^{2})+\beta_{WS}\sigma_{X-\bar{x}}^{2}/(\sigma_{X-\bar{x}}^{2}+\sigma_{\bar{x}}^{2})$$

In the previous example of weight and cholesterol with $\beta_{BS}$ = 0.9, $\beta_{WS}$ = 3 and $Y_{ij}=30+0.9{\bar{x}}_{i}+3(X_{ij}-{\bar{x}}_{i})+\varepsilon_{ij}$, if $\sigma_{\bar{x}}^{2}=400$ and $\sigma_{X-\bar{x}}^{2}=100$, then from (3) $\beta_{CS}$ = 0.9*400/(100+400)+3*100/(100+400) = 1.32. If the between-person sample means are more homogeneous in weight with $\sigma_{X-\bar{x}}^{2}=200$ but the within- person $\sigma_{X-\bar{x}}^{2}$ is still 100, then again using (3) $\beta_{CS}$ moves closer to $\beta_{WS}$; $\beta_{CS}$ = 0.9*200/(100+200)+3*100/(100+200) = 1.60. 

**2D Working Correlation Structures for Model Residuals Other than Independence Can Lead to Unusable Results for Cross-Sectional Regression.** As noted earlier, fitting both MM and GEE repeated measure regression models involves specification of correlation (or working correlation) structure of *ε_ij_* within the same subject *i*. We denote the working correlation structure by matrix *V_i_*. Typical choices for *V_i_* are the ones we used in the illustrative examples of Table 1 and Table 2; equicorrelation (E), with correlation of *ε_ij_* and *ε_ij_*_’_ for j ≠ j’ always the same value ρ (this common value of ρ is estimated in the model fitting process based on the residuals in the model fitting process), and independence (IND), with correlation of ε_ij_ and ε_ij’_ ≡ 0. However, other structures are used such as AR(1) where correlation of *ε_ij_* and *ε_ij’_* is *ρ^|j-j’|^* with the value of *ρ* being estimated from the residuals [1]. Again, current guidance [5,6,7,8,9,10,11,12,13] emphasizes choosing the *V_i_* that most closely fits the true covariance structure of the residuals within *i* and/or by model fit criteria such as having lowest QIC for GEE and AIC for MM, because doing so often improves precision of the model parameter estimates. However, we just observed that this approach may be wrong for CS regression, because using any correlation structure other than IND can introduce structural bias into ${\underset{˜}{\hat{\beta }}}_{CS}$ [20,21] and, unfortunately, AIC and QIC do not account for this bias. 

To that end, Pepe and Anderson (1994) [20], developed a general rule for when IND is (and is not) the only correlation structure that should be used for CS regression that we now present. Specifically, they show that if a predictor ${\underset{˜}{X}}_{ij}$ varies (i.e., takes on different values) within the same subject *i* and,
(4)$$E[Y|{\underset{˜}{X}}_{ij}]\;\mathrm{depends}\;\mathrm{on}\;X_{k,ij}\;\mathrm{for}\;\mathrm{any}\;k\;\mathrm{of}\;a\;\mathrm{different}\;\mathrm{replicate}\;j\prime \;\mathrm{in}\;i$$
then, no matter what true correlation structure of ε_ij_ among repeated measures within a subject is, GEE-IND gives unbiased estimates for ${\underset{˜}{\beta }}_{CS}$, but any MM or GEE model not using **V_i_** = IND, gives biased estimates of ${\underset{˜}{\beta }}_{CS}$. Thus, the only working correlation structure that should be used to estimate ${\underset{˜}{\beta }}_{CS}$ is **V_i_** = IND. However, if (4) does not hold, then any working correlation structure obtains unbiased estimates for ${\underset{˜}{\beta }}_{CS}$ in which case, choosing the **V_i_** that most accurately fits the correlation structure of ε_ij_ minimizes the variance of ${\hat{\underset{˜}{\beta }}}_{CS}$.

Our paper only focuses on equicorrelation as the alternate to independence in order to keep the presentation from becoming too cumbersome, given the large number of possible correlation structures. However, the previous paragraph and (4) apply to any non-independence correlation structure.

As one (of many) examples of where (4) holds, let k = 1 and *Y_ij_* and *X_ij_* be the degree of airway obstruction and inhalation of tobacco smoke of subject *i* at time *j*, respectively. One would expect that, because smoking effect on the lung is cumulative, historical smoking in a current smoker or non-smoker would lead to poorer lung function. Thus, *E[Y_ij_|X_ij’_]* for a smoker at time *j’* < *j* would poorer irrespective of *X_ij_*.

We now present an easier way to visualize (4). If repeated measures *j* and *j’* are thought of as “siblings” and the predictors as “exposures” then (4) means that even after considering the “self-exposure” of the current measure *j* through ${\underset{˜}{X}}_{ij}$ the outcome *Y* has “**Co**nditional **D**ependence **O**n **S**ibling **E**xposures” (Co-DOSE) (i.e., on $X_{k,ij\prime }$). Thus, the sibling exposure $X_{k,ij\prime }$ could be thought of as a Co-DOSE beyond the “dose” from the “self-exposure”. Hence, from now on we use the term Co-DOSE to denote that (4) occurs.

Also, while this point has not been very well made, for CS regression, Co-DOSE in (4) largely occurs if and only if within- and between-subject slopes differ. If within- and between-subject slopes differ *for any predictor* (i.e., ${\underset{˜}{\beta }}_{BS}\neq {\underset{˜}{\beta }}_{WS}$) then Co-DOSE (4) happens. However, if the within- and between-subject slopes are equal *for all predictors* (i.e., ${\underset{˜}{\beta }}_{BS}={\underset{˜}{\beta }}_{WS}$) then Co-DOSE (4) does not occur. More details on this and an illustration are given in Appendix B, but one trivial case arises if the predictors are invariant within the same subject (i.e., ${\underset{˜}{X}}_{i1}\equiv {\underset{˜}{X}}_{i2}\equiv ..\equiv {\underset{˜}{X}}_{iJ_{i}}\equiv {\underset{˜}{\bar{x}}}_{i}$) such that the within-subject slopes are not defined (since ${\underset{˜}{X}}_{ij}-{\underset{˜}{\bar{x}}}_{i}\equiv 0$) and for the same reason Co-DOSE in (4) cannot occur. While the mathematical details are beyond this paper, if ${\underset{˜}{\beta }}_{BS}\neq {\underset{˜}{\beta }}_{WS}$ and ***V_i_*** = IND, then non-zero covariance *ρ_ij_* > 0 besides adjusting for within-*i* collinearity of *ε_ij_ also* over-weights the ${\underset{˜}{\beta }}_{WS}$ relative to ${\underset{˜}{\beta }}_{BS}$ in (3), thereby pushing CS regression parameter estimates away from ${\underset{˜}{\beta }}_{CS}$ towards ${\underset{˜}{\beta }}_{WS}$ [17]. Since robust covariance methods exist to adjust for impact of misspecification of ***V_i_*** = IND from collinearity of the residuals *ε_ij_’s* on variance estimates, in particular for GEE [1], ***V_i_*** = IND can eliminate bias in estimating ${\underset{˜}{\beta }}_{CS}$ while providing conservative variances for the parameter estimates.

**2E Implications for Applied Research and Statistical Practice.** Much of what has been presented above is not commonly understood and implemented in applied research and statistical practice. CS models are typically fit, with ${\underset{˜}{\beta }}_{CS}$ interpreted to also be ${\underset{˜}{\beta }}_{BS}$ and ${\underset{˜}{\beta }}_{WS}$, without checking if these slopes are equal. Non-independence***V_i_*** is often used for CS regression without checking if Co-DOSE (in (4)) exists. Perhaps in part this occurs because systematic epidemiological descriptions of causal mechanisms for why between- and within-subject slopes can differ are lacking, which hinders awareness of this possibility. We endeavor to fill this gap in Section 3. 

## 3. Epidemiological Reasons for Between- and Within-Subject Slopes to Differ

To make it easier for investigators to identify what could cause *β_k,WS_* ≠ *β_k,BS_* (or equivalently Co-DOSE) in a given setting, we classify major reasons why this can happen. For simplicity, let *K* = 1 unless otherwise noted, as the following principles extend to multivariate settings.

**3A. Change Effects.** We propose that the effect of a longitudinal within-subject change in the predictor X could have a greater (or less) direct impact on Y than a long-term standing difference in X between two different subjects (hence *β_WS_* ≠ *β_BS_*) and define this as a (c.f. short term) “change effect”. Returning to the example of weight and cholesterol, consider two identical twins, “A” has lived his adult life at ${\bar{x}}_{i}$ = 190 lbs. and “B” at ${\bar{x}}_{i\prime }$ = 180 lbs. If “B” undergoes a short-term weight gain of 10 lbs. to 190 ($X_{i\prime j}-{\bar{x}}_{i\prime }$ = 10), assuming ${\bar{x}}_{i\prime }$ not impacted by the rapid change, while A remains at 190 lbs. ($X_{ij}-{\bar{x}}_{i}$ = 0), the shock or corollaries of this rapid change in B may raise his cholesterol level above that of A’s even though they both now weigh 190 lbs., meaning that *β_WS_* > *β_BS_* and Co-DOSE in (4) occurs. However, it should be noted that as was mentioned in Section 2B, in this setting, *β_WS_* would be somewhat undefined if, e.g., a 10 lbs. gain in a shorter time period (i.e., 1 month) increases *β_WS_* more than does a 10 lbs. gain over a longer time period (i.e., 12 months).

**3B Lag Causality of *X* on Future *Y*.** The effect of historical levels of *X* on *Y* may independently project into the future (i.e., beyond that effect of the current level of X). For example, consider an HIV-infected person and two time points *t_1_* < *t_2_*; let *X_ij_* be HIV viral load and *Y_ij_* be CD4 count. High HIV levels destroy CD4 blood cells into the future. Therefore, as illustrated in Figure 2a, high HIV viral load at *t_1_* may affect CD4 loss from t_1_ to t_2_ so that even if the person’s HIV viral load is low at t_2_, the high viral load at t_1_ is predictive of lower CD4 at t_2_ through higher viral load at t_1_ having created more CD4 destruction between t_1_ and t_2_ (i.e., lag causality of X at t_1_ on Y at t_2_). Thus, *Y_i2_|X_i2_* at *t_2_* is not independent of *X_i1_* at t_1_; Co-DOSE in (4) occurs and the within- and between- subject slopes differ (*β_WS_* ≠ *β_BS_*). In Figure 2a,b, *E_i2_* denotes that *X_i2_* differs from *X_i1_* due to an extraneous process that is causing *X_i_* to change over time. Lag causality is often considered when serial measures of *X* represent long-term environmental exposures (such as air pollution and cigarette smoke) that effect chronic conditions *Y* (such as lung function) are obtained [1,18,22].

**3C Reverse-Lag Causality of *X* on Future *Y.*** The setting in Section 3B also manifests in the opposite direction if *X* is being used as to estimate *Y* that is causal for future *X*. Reversing the previous example with *X* now being CD4 used to predict HIV viral load as Y, as Figure 2b illustrates, high viral load (*Y_i1_*) at t_1_ may have degraded the CD4 count from t_1_ to t_2_. Thus, *Y_i1_|X_i1_* at *t_1_* is not independent of *X_i2_* at *t_2_*: Co-DOSE in (4) occurs and within- and between-subject slopes differ (*β_WS_* ≠ *β_BS_*). 

**3D Spillover Causality of X on Adjacent Y.** An analogous setting to those of 3B and 3C can also manifest in repeated measure cross-sectional settings based on geographical proximities. Let the subjects *i* now be cities and *j* enumerate different neighborhoods in these cities. The repeated measures are average air pollution (*X_ij_*) of neighborhood *j* in city *i* and average lung function of all residents living within neighborhood *j* of city *i* (*Y_ij_*). A resident living in neighborhood *j* may work in a different neighborhood *j’* of the same city and thus have “*spillover exposure*” to air in the neighborhood they work in, for a given city i, thus *Y_ij_|X_ij_* is not independent of *X_ij’_* and hence Co-DOSE in (4) occurs.

**3E Common Within-Subject Measurement Bias.** Shared *within-subject measurement bias* occurs if all repeat measures from the same subject have the same correlated measurement bias. This was the setting described in Section 2B and Figure 1a with weight as exposure for cholesterol. Here with weight as a surrogate for body fat, the measurement bias was mediated by height with taller adults being heavier independently of body fat than shorter adults, which leads to *β_WS_* > *β_CS_* and Co-DOSE in (4) when weight was a predictor of cholesterol. In this setting, height is a *measurement bias* not a confounder as height itself is not associated with cholesterol. We now present a similar setting where the un-modeled variable is a confounder. 

**3F Common Within-Subject Confounding.**
Figure 1b shows *common within-subject confounding*, that causes *β_WS_* ≠ *β_CS_* and Co-DOSE in (4). This phenomenon is diagrammatically similar to *common measurement bias* that was described in Section 2B. However, rather than a common measurement bias, the extraneous factor, shared by the repeated measures of the same subject, is a confounder that is associated with both X and Y. For example, let the confounder variable *C_i_* be sex of subject *i* (which does not change with j) not be in the model and the outcome *Y_ij_* be a linear score for male pattern baldness at time j with again *X_ij_* being weight at time j. Adult men are both on average heavier and, independently of weight, have greater male pattern baldness than do adult women. So *C_i_* is associated with both the exposure and the outcome. Here a 10 lbs. weight difference in two adults, but not a within-adult increase of 10 lbs., could be informative of the heavier adult more likely being male. Hence for this example, *β_WS_* = 0 (assuming within adult weight does not influence baldness), but *β_CS_* > 0 (and thus *β_BS_* > 0) as males are more likely to be both heavier and bald compared to women. Hence also *β_CS_* > 0, reflecting unaddressed between-subject confounding from heavier adults more likely being men. 

Similarly, Mancl, Leroux and DeRouen proposed that in a study with repeated dental predictor and outcome pairs as (*X_ij_,Y_ij_*) measured on teeth (i.e., enumerated by *j*) on the same persons (i.e., enumerated by *i*) that better compliance with dental treatment by some persons was a confounder that could lead to differences in slopes within and between subjects [19]. In a non-longitudinal setting where *i* denotes clusters (for example schools) and *j* denotes repeated subjects within that cluster (for example students), common within-subject confounding is referred to as “contextual effects” [23,24]. For example, as Robinson (1950) [14] observed, when X was race of the student and *Y* was achievement-score, a higher ${\bar{x}}_{i}$ (here: portion of a school’s students that were non-White) indicated weaker financial support for that school (weaker financial support being the confounder) and thus worse achievement-scores overall for that school: *β_BS_* was negative. However, within the same school, race had no impact on the achievement score (*β_WS_* = 0). Begg and Parides [25] identify a similar setting in birthweight and intelligence quotient in families.

**3G Measurement Error in *X_ij_* Makes E[*Y_ij_|X_ij_*] Dependent on *X_ij’_*** In many settings, the predictor we observe is *X* = *TX* + *M* where *TX* is the true value of the predictor and *M* is measurement error that is independent of *TX* (i.e., classical measurement error). It has been shown that, measurement error in *X* that is either independent of [26], or correlated with *Y* [27], biases estimates for the slope that relates *TX* with *Y*. Measurement error can arise either from imprecision in an analysis instrument, such as in a machine quantifying components of serum, or *in data collection process,* such as the chemical composition of blood samples being non-informatively influenced by diurnal and other nuisance processes. If *X_ij_* is incorrectly quantified due to such measurement error, then Co-DOSE in (4) occurs and the observed within- and between-subject slopes differ, because, as illustrated in Appendix C, the biases being created from the measurement error distribute differentially to different slopes. As Figure 3 shows and the paragraph below it describes using an illustrative example, if *X_i1_* incompletely measures the true state *TX_i1_* (i.e., true BUN) due to classical measurement error as the extraneous influence then *X_i2,_* is informative for *TX_i1_* even after considering *X_i1_*. Please note that in Figure 3 there are two times subscripts on the extraneous influence, because *E_i1_* and *E_i2_* are two independent measure errors. 

For example, going back to the analysis of Table 1, let *X_ij_* be BUN and *Y_ij_* be EGFR. Consider two persons who have BUN of *X_i1_* = 10 mg/dL measured with error today. Also assume that the true BUN state changes slowly. If so, and after 6 months one of these persons measures *X_i2_* = 20 mg/dL while the other measures *X_i2_* = 5 mg/dL, we can then surmise that since BUN changes slowly, it is more likely that the true BUN today (*TX_i1_*) of the former person is > 10 mg/dL and that of the latter is < 10 mg/dL. Thus, since (i) EGFR (*Y_i1_*) directly depends on *TX_i1_* not *X_i1_*, and (ii) *X_i2_* is informative on *TX_i1_* after considering *X_i1_*, then (iii) *Y_i1_|X_i1_* is not independent of *X_i2_* and similarly *Y_i2_|X_i2_* not independent of *X_i1_* meaning Co-DOSE in (4) occurs and the observed within- between-subject slopes differ. Appendix C shows that measurement error in the exposure that is independent of the outcome pushes both *β_WS_* and *β_BS_* towards 0, but more so for β_WS_. Such tempering from averaged measurement error has been proposed as a reason |*β_WS_*| < |*β_BS_*| was observed in dental research [19] and occupational epidemiology [28,29]. 

However, if *M_ij_* is correlated with *Y_ij_* (most likely being correlated with measurement error on *Y_ij_* [27]) the tempering of *β’s* from *M_ij_* will not be to 0. For example, consider *TX* = CD8 and *TY* = CD4 cells which together are the almost exclusive components of serum lymphocytes (*TZ*) (i.e., TY≈TZ−TX. Physiologically, *TZ* is constrained to create a negative *β_BS_*, *β_WS_* and *β_CS_* for *TY_ij_* on *TX_ij_*: subjects with a higher CD8 component of serum lymphocytes by converse must a have lower CD4 components. However, the measured lymphocyte count (*Z*) is subject to a correlated measurement error that equally spreads onto *X* and *Y*. For example, if a person is dehydrated, the entire measured lymphocyte (meaning both CD8 = X and CD4 = Y) portion of blood becomes artificially higher due to reduction of the percentage of water in the blood. If a person has a high (or low) measured lymphocyte count *Z_ij_* = *TZ_ij_* + *M_ij_* due to such measurement error, then *M_ij_* contributes to both CD4 (*X_ij_*) and CD8 (*Y_ij_*), making both simultaneously artificially higher (or lower). Consequently, within person, a higher measured CD4 count due to positive *M_ij_* is associated with higher measured CD8. Because in this case the measurement error is shared, naïve regression analysis tends to draw *β_WS_* towards being positive. On the other hand, *β_BS_,* which tempers down *M_ij_* on both *X* and *Y* through averaging as shown in Appendix C, is less affected by the shared bias due to measurement error.

We have only considered classical measurement error so far. The other common type of measurement error is known as Berkson error [30]. It is approximated by some exposure assessment procedures commonly used in environmental and occupational epidemiology (see semi-ecological design and group-based exposure assessment) [18]. While this is an aside to the main points of this paper, when Berkson measurement error exists, only the between-subject slope, *β_BS_*, is estimable. More details are in Appendix D.

## 4. Predictors Having Co-DOSE Will Bias Adjusted Parameter Estimates of Other Predictors Not Having Co-DOSE When Included Together in Cross-Sectional Regression When *V_i_* ≠ IND Is Used

Going back to Table 1, it was shown earlier that the point estimate from GEE-IND ${\hat{\beta }}_{HIV,CS}$ for the adjusted cross-sectional association of HIV with EGFR is still consistent for $\beta_{HIV,CS}$. However, HIV infection status was constant over all replicates within the same subject, and therefore cannot have Co-DOSE in (4) as the entire effect of HIV is mediated between-subject, not within-subject. Consequently, the question arises whether the adjusted estimate from a non-independence correlation structure (say for example ${\hat{\beta }}_{HIV,CS-E}$) can be biased for $\beta_{HIV,CS}$. Please note that for this section, we use ${\hat{\beta }}_{XXX,CS}$ and ${\hat{\beta }}_{XXX,CS-E}$ to denote estimates for adjusted cross-sectional association for variable XXX from models using independence and equicorrelation structures, respectively. The added designation of “E” (CS-E) in the subscript for equicorrelation, but none for independence correlation, is made because the equicorrelation estimate (but not the independence estimate) can be asymptotically biased. The specific question addressed here is: could including BUN and albumin that each have Co-DOSE in the model bias the corresponding estimate for cross-sectional adjusted HIV association from using equicorrelation (${\hat{\beta }}_{HIV,CS-E}$) so that it no longer is consistent for β_HIV,CS_ in the multivariate model, even though HIV itself is not Co-DOSE? This is important, because in Table 1, ${\hat{\beta }}_{HIV,CS}$ of −2.04 95% CI (−5.07 0.98) qualitatively differs from ${\hat{\beta }}_{HIV,CS-E}$ of −3.96 (−6.90, −1.03) with only ${\hat{\beta }}_{HIV,CS-E}$ statistically (*p* < 0.01) differing from 0.

We believe that ${\hat{\beta }}_{HIV,CS-E}$ for HIV in Table 1 is biased away from *β_HIV,CS_*. To help make this point, Table 3 presents normative data broken down by HIV status of the subjects. First we note from Table 1 that ${\hat{\beta }}_{BUN,CS-E}$ is biased higher for $\beta_{BUN,CS}$ (with GEE-E ${\hat{\beta }}_{BUN,CS-E}$ = −1.22 > ${\hat{\beta }}_{BUN,CS}$ = −1.87, *p* < 0.0001 from GEE-IND), while from Table 3, those who are HIV+ have higher mean BUN (12.94 vs. 12.10, *p* < 0.0001 from GEE-IND). Thus, the full apparent “negative effect” of the higher BUN in HIV+ subjects from $\beta_{BUN,CS}$ is underestimated by ${\hat{\beta }}_{BUN,CS-E}$ and this pushes ${\hat{\beta }}_{HIV,CS-E}$ down to compensate. Second, similarly, also from Table 1, ${\hat{\beta }}_{ALB,CS-E}$ is biased lower for *β_ALB,CS_* (with GEE-E ${\hat{\beta }}_{ALB,CS-E}$ = −9.84 < ${\hat{\beta }}_{ALB,CS}$ = −6.21), while from Table 3, HIV+ individuals have lower mean albumin (3.97 vs. 4.14, *p* < 0.0001 from GEE-IND). Thus, the apparent “positive effect” of the lower albumin in HIV+ subjects from $\beta_{ALB,CS}$ is overestimated by ${\hat{\beta }}_{ALB,CS-E}$, which pushes ${\hat{\beta }}_{HIV,CS-E}$ further down to compensate. Now we consider these two biases together as illustrated in Figure 4. These two deficits act jointly to push ${\hat{\beta }}_{HIV,CS-E}$ downwards from the true adjusted *β_HIV,CS_*. Therefore, non-independence ***V_i_*** can bias multivariate cross-sectional parameter estimates of variables that do not carry Co-DOSE in (4) when other variables in the model carry Co-DOSE.

## 5. Discussion

Numerous published papers fit GEE and MM cross-sectional regression models with repeated measures having time varying predictors that either use non-independence working correlations structures or do not state the correlation structure. These papers, which continue to be published, do not show awareness of the points presented in Section 1, Section 2, Section 3 and Section 4, above. Specifically, they:(a)Neither specify whether the coefficients of interest are ${\underset{˜}{\beta }}_{CS}$, ${\underset{˜}{\beta }}_{WS}$ or, ${\underset{˜}{\beta }}_{BS}$ nor check whether ${\underset{˜}{\beta }}_{WS}={\underset{˜}{\beta }}_{BS}$;(b)Make potentially invalid interpretations of ${\underset{˜}{\beta }}_{CS}$ from MM and GEE using non-independence correlation ***V_i_’s***; and/or; (c)Do not justify the choice of non-independence working correlation structures ***V_i_*** in light of potential differences between ${\underset{˜}{\beta }}_{WS}$, ${\underset{˜}{\beta }}_{BS}$ and ${\underset{˜}{\beta }}_{CS}$.

We have identified almost 45 such papers including some authored by us prior to becoming aware of these issues. This is almost certainly only a fraction of the total number of such papers.

Yet papers published up to 65 years ago either warn against using non-independence working correlation structure in cross-sectional regression with repeated measures [19,20], or instruct to decompose the associations into within-subject (${\underset{˜}{\beta }}_{WS}$) and between-subject (${\underset{˜}{\beta }}_{BS}$) slopes to make causal inference [14,15,16,17]. Numerous examples where ${\underset{˜}{\beta }}_{WS}\neq {\underset{˜}{\beta }}_{BS}\neq {\underset{˜}{\beta }}_{CS}$ have been presented [14,15,16,17,18,19,20,22,23,24,25]. While it was not covered in our paper, this includes fitting GEE models of binary outcomes where the issues discussed here also apply [19,31]. However, these points are still not well known or emphasized in statistical software documentation and papers providing guidance on GEE and MM analyses (i.e., [5,6,7,8,9,10,11,12,13]).

One problem that impedes acceptance of within- and between-subject decomposition is that it necessitates much more complicated models that are difficult to explain. Still, some air pollution epidemiologic studies have attempted within- and between-subject decompositions using cities as the subject and neighborhoods as the repeated measures within the city [32,33,34]. Most often in these studies, the magnitude was greater for within-subject slope |β_WS_| > |β_BS_| but sometimes |β_BS_| > |β_WS_| was observed meaning that possibly multiple causes for slope differences are involved. Those papers that did attempt to explain the reasons for the differences described only “common within-subject confounding” (Section 3E) as a potential reason; such as un-modeled pollutants that were correlated between (but not within) cities with the modeled pollutants of interest. Other studies in environmental research have considered the mechanism described in Section 3B, namely, lag causality in longitudinal analyses of association of air pollution on health measures [1]. Nevertheless, having to explain complicated and unknown mechanisms for biases such as these can appear to detract from the main purpose of the research and cast doubt on the overall findings, making the paper harder to publish. In other words, there appears to be neither incentive, nor guidance on how to engage with these issues for applied researchers.

We concur with others [19,20], that cross-sectional regression with repeated measures should use independence as the default working correlation unless justification is given to use other ***V_i_***. While non-independence ***V_i_*** can improve precision and thus be desirable [21], they can considerably bias estimates for cross-sectional parameters, ${\underset{˜}{\beta }}_{CS}$, including perhaps towards what the investigator wants to see. For example, in Table 1, *p* < 0.01 was observed for association with HIV with worse EGFR in GEE-E compared to the more appropriate *p* = 0.19 from GEE-IND. An investigator who was expecting HIV to be associated with worse EGFR might thus be tempted to use the results from GEE-E for this reason.

While showing this is beyond the scope of our paper, when ***V_i_*** is not independence, factors such as the values of *J_i_* and magnitude/structure of *ε_ij_* strongly influence parameter estimate values for ${\underset{˜}{\beta }}_{CS}$ from the miss-fitted cross-sectional models, allowing the miss-fitted estimate to arbitrarily range from ${\underset{˜}{\beta }}_{CS}$ to ${\underset{˜}{\beta }}_{WS}$ [17]. Standardization is important and, as such factors will arbitrarily vary between studies, parameter estimates of ${\underset{˜}{\beta }}_{CS}$ become harder to compare across studies when ***V_i_*** differs at discretion of investigators. Therefore, the working correlation structure used in cross-sectional regressions using repeated measures should always be justified and reported.

We also concur with others [14,15,16,17,18,19,23,24,25] that despite the difficulties in identifying why within- and between-subject slopes differ, causal inference analyses with repeated measures should initially make such decompositions. Investigators should then be wary if there are qualitative differences between ${\underset{˜}{\beta }}_{WS}$ and ${\underset{˜}{\beta }}_{BS}$. For example, Table 2 with 584 subjects and 10,782 measurements demonstrated need for ${\underset{˜}{\beta }}_{WS}$, ${\underset{˜}{\beta }}_{BS}$ decomposition to make causal inference (as well as for using GEE-IND in cross-sectional regression). However, a smaller study could have been less clear-cut. If the same point estimates for ${\underset{˜}{\beta }}_{WS}$ and ${\underset{˜}{\beta }}_{BS}$ seen in Table 2 were observed but did not statistically differ, one would be tempted to merge ${\underset{˜}{\beta }}_{WS}$ and ${\underset{˜}{\beta }}_{BS}$ into a combined ${\underset{˜}{\beta }}_{CS}$ at least for some variables, because standard model-fitting practice promotes parsimony when statistical significance is not observed. This would be particularly true if for a given variable, *k*, neither ${\hat{\beta }}_{k,WS}$ nor ${\hat{\beta }}_{k,BS}$ statistically differed from 0, but ${\hat{\beta }}_{k,CS}$ did. If such collapsing is done, it may still be important to report ${\hat{\beta }}_{k,WS}$ and ${\hat{\beta }}_{k,BS}$ for comparison to future studies and target potential mechanisms for within- between-subject slope differences as described in Section 3.

Unfortunately, the within- and between-subject slope decomposition expands required analyses and presentation. Statistical software mostly does not have standard subroutines to do this. Decomposition can be tedious if ${\bar{x}}_{k,i}$ is recalculated to maintain orthogonal decomposition of *X_k,ij_* as new models are fit if observations are excluded from the *J_i_* due to missing values of newly included variables. The fact that the ${\bar{x}}_{k,i}$ are ill-defined by averaging the *X_k,ij_* rather than being true means for subject *i* creates confusion about interpretation of ${\hat{\underset{˜}{\beta }}}_{BS}$ that can also be influenced by within-subject slopes as was noted in Section 2B.

When ${\underset{˜}{\hat{\beta }}}_{BS}$ and ${\underset{˜}{\hat{\beta }}}_{WS}$ differ, the causal mechanisms as to why this happens should be explored. For example, in our analysis presented in Table 2 with EGFR as the outcome, for BUN the between-subject slope ${\hat{\beta }}_{BUN,BS}=-2.72$ (from GEE-IND) was statistically further from 0 in the expected direction of association than was the within-subject slope ${\hat{\beta }}_{BUN,WS}=-1.11$. However, the albumin went the other way: between-subject slope ${\hat{\beta }}_{ALB,BS}=-3.27$ was statistically closer to 0 than was within-subject slope ${\hat{\beta }}_{ALB,WS}=-10.70$ with again both slopes being in the expected direction from zero. So what are the potential reasons for this? While lag/reverse-lag causality (Section 3B,C) between BUN and creatinine (the main component of calculated EGFR) could reduce magnitude of β_BUN,WS_ vs. β_BUN,BS_, this was unlikely given the separation of visits was 6 months and internal biochemistry operates over shorter time periods. However, independent measurement error on BUN (Section 3G) would temper |β_BUN,WS_| towards 0 relative to |β_BUN,BS_|. To that end, several articles find greater coefficient of variation [35,36], within-person change [35,36], assay error [36], and sample degradation for BUN vs. albumin measures [37], all of which could reflect BUN having larger independent measurement error than does albumin that would selectively attenuate ${\hat{\beta }}_{BUN,WS}$ towards 0 (i.e., more than it did to ${\hat{\beta }}_{ALB,WS}$). Conversely, serum creatinine and albumin are both constrained into the intravascular fluid compartment and will non-informatively increase together with greater hydration and decrease with less hydration of this compartment, inducing positively correlated measurement error, as in the case for measured CD4 and CD8 cells in the last paragraph of Section 3G. As creatinine factors inversely into the EGFR calculation, this would constitute negative correlation of measurement error between albumin and EGFR and selectively bias ${\hat{\beta }}_{ALB,WS}\;$ to be more negative than ${\hat{\beta }}_{ALB,BS}\;$. However, BUN, which crosses across all body compartments, is less subject to such correlation in measurement error with creatinine and thus with EGFR.

As is illustrated in the previous paragraph, we believe that the systematic epidemiological description of reasons for within- and between subject slopes to differ in Section 3 will provide some basis for future studies to explore this. That may lead to greater recognition and understanding of this phenomenon. However, our list of reasons for these slopes to differ may not be exhaustive. Furthermore, these mechanisms are quite complicated including that limited resources may be available to investigate them in given studies given the other tasks that need to be done and limited funding/personnel.

When between- and within-subject slopes differ, ${\underset{˜}{\beta }}_{BS}\neq {\underset{˜}{\beta }}_{WS}$, it is unclear which is the “least confounded or biased”, including the possibility that by “averaging” the different biases in each would make ${\underset{˜}{\beta }}_{CS}$ be the least biased. There may be a heuristic perception that by “matching within the same subject”, ${\underset{˜}{\beta }}_{WS}$ is superior to ${\underset{˜}{\beta }}_{BS}$ and ${\underset{˜}{\beta }}_{CS}$, but this is not necessarily true as measurement error in X (Section 3G) and lag/reverse-lag and spillover causality (Section 3B–D) can in fact bias ${\underset{˜}{\beta }}_{WS}$ to a larger degree than they do for ${\underset{˜}{\beta }}_{BS}$ and ${\underset{˜}{\beta }}_{CS}$.

## 6. Conclusions

It has been known for decades by some that when exposures vary within subjects in repeated measures regression then, (i) cross-sectional regression using ***V_i_*** = independence working correlation should be the default for building a model to estimate a future unknown *Y* as the goal, and (ii) within- and between-subject decompositions of slopes should at least initially be fit when building models for causal inference. Yet this advice rarely makes it into published guidelines and hence is not heeded, perhaps in part due to complexity of the settings where within- and between-subject slopes differ and limited substantive study of the mechanisms that cause such differences. In general, analysts should explore and quantify reasons for biases that can occur in such study designs. To that end, analyses using repeated measures regression should investigate if within- and between-subject slopes differ and when they do, try to identify the reasons for this.

## Figures and Tables

**Figure 1 ijerph-16-00504-f001:**
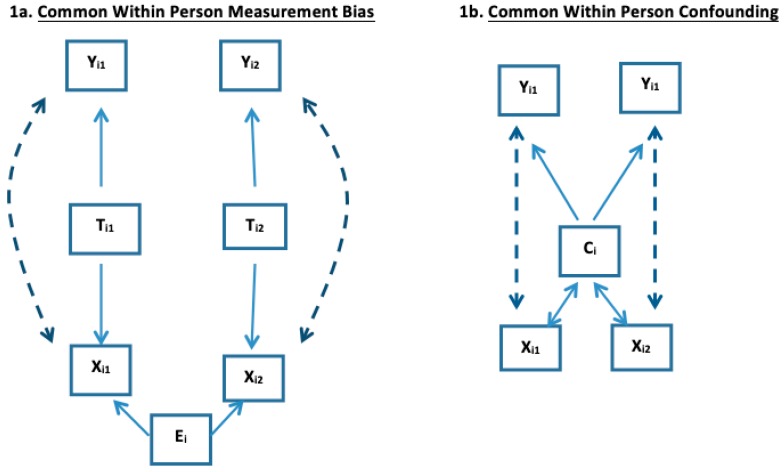
Illustration of common within-subject measurement bias and confounding for K = 1.

**Figure 2 ijerph-16-00504-f002:**
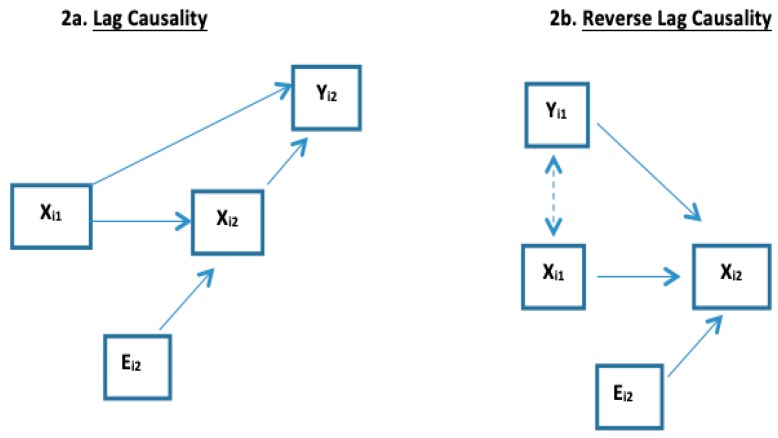
Illustration of lag causality and reverse-lag causality for K = 1.

**Figure 3 ijerph-16-00504-f003:**
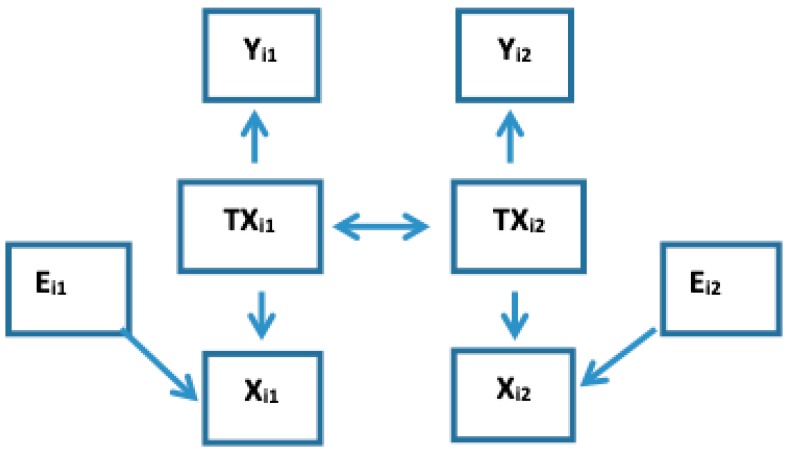
Illustration of residual association with independent measure error in X for K = 1.

**Figure 4 ijerph-16-00504-f004:**
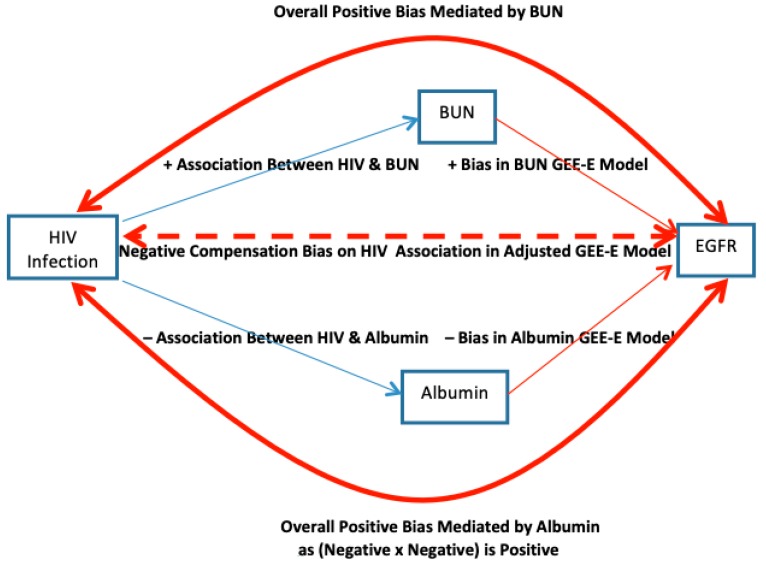
Compensating bias pathways on time invariant HIV estimate from failure to use an independent working correlation structure in repeated measures GEE.

**Table 1 ijerph-16-00504-t001:** Cross-sectional regression parameter estimates using GEE ^1^ for EGFR = HIV infection, serum albumin and BUN in the Bronx WIHS.

Variable	Working Correlation Structure					
	Independence			Equicorrelation ^2^		
	Point Estimate	95% CI	Z-Value (*p*)	Point Estimate	95% CI	Z-Value (*p*)
HIV Infection (β_HIV,CS_)	−2.04	(−5.07, 0.98)	−1.32 (<0.19)	−3.96	(−6.90, −1.03)	−2.65 (0.0081)
Albumin Per g/dL (β_ALB,CS_)	−6.21	(−8.95, −3.47)	−4.44 (<0.0001)	−9.84	(−12.01, −7.68)	−8.93 (<0.0001)
BUN Per mg/dL (β_BUN, CS_)	−1.87	(−2.12, −1.62)	−14.45 (<0.0001)	−1.22	(−1.46, −0.99)	−10.30 (<0.0001)
Quasi-Likelihood Information Criteria (QIC)	10,847.14			10,836.27		

^1^ Mixed models gave essentially similar point estimates; see Appendix A. ^2^ Interclass correlation of residuals from GEE-E was 0.45 indicating non-independence correlation was structurally correct.

**Table 2 ijerph-16-00504-t002:** Within- and between-subject decomposition regression parameter estimates using GEE ^1^ for EGFR = HIV infection, serum albumin and BUN in the Bronx WIHS.

Variable		Working Correlation Structure					
	Compartment	Independence			Equicorrelation		
		Point Estimate	95% CI	Z-Value (*p*)	Point Estimate	95% CI	Z-Value (*p*)
HIV Infection	Between-subject (β_HIV, BS_)	−1.16	(−4.21, 1.88)	−0.75 (0.45)	−1.57	(−4.47, 1.33)	−1.06 (0.29)
	NA ^2^	---	---	---	NA ^2^	---	---
Albumin Per g/dL	Between-subject (β_ALB, BS_)	−3.27	(−7.88, 1.33)	−1.39 (0.16)	−2.71	(−7.00, 1.57)	−1.24 (0.21)
	Within-subject (β_ALB, WS_)	−10.70	(−12.99, −8.40)	−9.16 (<0.0001)	−10.70	(−12.99, −8.40)	−9.16 (<0.0001)
BUN Per mg/dL	Between-subject (β_BUN, BS_)	−2.72	(−3.10, −2.33)	−13.89 (<0.0001)	−2.65	(−3.01, −2.08)	−14.21 (<0.0001)
	Within-subject (β_BUN, WS_)	−1.11	(−1.34, −0.88)	−9.31 (<0.0001)	−1.11	(−1.34, −0.88)	−9.31 (<0.0001)
Quasi-Likelihood Information Criteria (QIC)		10,866.64			10,857.62		

^1^ Mixed models gave essentially similar point estimates. See Appendix A
^2^ There is no within-subject variation for HIV infection status.

**Table 3 ijerph-16-00504-t003:** Means ± standard deviation of EGFR serum albumin and BUN broken down by HIV status across all repeated measures used in Table 1 and Table 2.

Variable	For HIV + Subjects (496 persons 7326 Replicates)	For HIV - Subjects (178 persons 3456 Replicates)
**EGFR**	90.3 ± 27.2	92.4 ± 25.0
**BUN**	12.94 ± 5.71	12.10 ± 5.30
**Albumin**	3.97 ± 0.44	4.14 ± 0.36

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
