# Peer review of "Repeated Measures Regression in Laboratory, Clinical and Environmental Research: Common Misconceptions in the Matter of Different Within- and Between-Subject Slopes"

_ijerph, 2019, doi:10.3390/ijerph16030504_

Round 1
Reviewer 1 Report
This is a quite interesting topic but the manuscript is not clear at all. I would say that it is very difficult to read and therein lies the main issue with this manuscript. The language used is very confusing and I (and a colleague who I asked to read parts of it) cannot really follow it at places. For example, in the abstract the example they give at lines:20-24 is very difficult to follow, sounds more like something google stitched together. In following sections where statistical models come into play, things become even more convoluted. For me, section 2 was very hard to follow and frankly I did not follow all of it, and thus I think that for a non specialist it will be impossible to follow. The authors have to re-write the manuscript keeping in mind that it has to be read and understood by (at least) their colleaques and more likely non statisticians who need to use mixed models and GEE's. The manuscript would also benefit from exact specification of models, perhaps supplemented by code in a statistical language/package that can be deciphered by the user. For example (my comment is not limited to this though): for the mixed model, I have no idea if they are refering to a random intercept only model or something else. I also believe that the correlation they are talking about in mixed models is for the residuals (they should say so) rather than the random effects.
A more specific question: what does this paper add above what reference 19 has added?
I would aslo use simulated data (which are easy to produce) where I would know the ubnderlying correlation structure and demonstrate the point using these rather than a real dataset for which i do not really know the underlying correlation structure. I would also give the code for this as an appendix
Thus, at the current point, I think that the manuscript is certainly not publishable mainly due to the language and flow which is basically (at least) very hard if not impossible to follow.
Author Response
SECOND REVIEW FROM REVIEWER ONE
I think that the authors have done enough for their paper to have a borderline acceptable readability for the general audience. I disagree with the authors that because it is a very technical it is acceptable to be too hard for thre average epi person. I have read highly technical issues explained very well. As a suggestion I would direct them to books published by Zuur et al. However, I do nto think there is any merit in insisting on this any more and i believe that the scientific community would benefit from the publication of this paper therefore I am suggesting the acceptance of it.
RESPONSE
We greatly appreciate your time and helpful suggestions, as well as, your confidence in the paper. We have gone back over the paper once again to improve grammar, shorten sentences and to rewrite in other ways to make things clearer. Yes. we agree that the work by Zuur is very good, for example https://besjournals.onlinelibrary.wiley.com/doi/10.1111/j.2041-210X.2009.00001.x. We hope and believe that once the methodology we advocate will gain acceptance and understanding among consulting statisticians and more analytical applied investigators, we can follow this up with a more accessible tutorial for non-statisticians and less analytical investigators along the lines of approach by Zuur et al.

Reviewer 2 Report
Thanks for the paper on “Repeated Measures Regression in Laboratory, Clinical and Environmental Research: Common Misconceptions in the Matter of Different within- and between- Subject Slopes.” It’s an important subject, especially when analysts struggle to choose different models when analyzing repeated measures. However, there are these few comments that the authors need consider:
1. Why did you choose only independence and equicorrelation structures and not even a random matrix to explain your point, yet we know that the random assumption is well placed in these researches? Please explain and add this point in the paper.
2. Page 9 Line 280: …we classify major reasons WHY this can happen????
3. As much as you focused on the slope, the role of fixed and random intercept on the slope needs to be mentioned briefly.
4. It would also be important to provide the results of the mixed models, even if it is a supplementary file.
5. What is the other important contribution of your study, if not the reasons why between and within-subject slopes differ that have also been identified by others?
6. Any limitations of your study? Could there be biases associated with authors opinion?
Author Response

(The authors gave the same response as above.)

Round 2
Reviewer 1 Report
I think that the authors have done enough for their paper to have a borderline acceptable readability for the general audience. I disagree with the authors that because it is a very technical it is acceptable to be too hard for thre average epi person. I have read highly technical issues explained very well. As a suggestion I would direct them to books published by Zuur et al. However, I do nto think there is any merit in insisting on this any more and i believe that the scientific community would benefit from the publication of this paper therefore I am suggesting the acceptance of it.
Author Response

(The authors gave the same response as above.)
